# Asymmetric Cascade Networks for Focal Bone Lesion Prediction in Multiple Myeloma

**Roxane Licandro**[1,2]                                ROXANE.LICANDRO@MEDUNIWIEN.AC.AT
[1] *Computer Vision Lab - Department of Visual Computing and Human-centered Technologies, TU Wien, Austria*

**Johannes Hofmanninger**[2]**, Matthias Perkonigg**[2]**, Sebastian Roehrich**[2]
[2] *Computational Imaging Research Lab - Department of Biomedical Imaging and Image-guided Therapy, Medical University of Vienna, Austria*

**Marc-André Weber**[3]
[3] *Institute f. Diagn. u. Intervent. Radiologie, Medizinische Universität Rostock, Germany*

**Markus Wennmann**[4]**, Laurent Kintzele**[4]
[4] *Department of Interventional and Diagnostic Radiology - Section Musco-Skelatal Imaging, Heidelberg University, Germany*

**Marie Piraud**[5]**, Bjoern Menze**[5]
[5] *Institute of Biomedical Engineering - Image-based biomedical modelling, Technische Universität München, Germany*

**Georg Langs**[2]                                        GEORG.LANGS@MEDUNIWIEN.AC.AT

## Abstract

The reliable and timely stratification of bone lesion evolution risk in smoldering Multiple Myeloma plays an important role in identifying prime markers of the disease's advance and in improving the patients' outcome. In this work we provide an asymmetric cascade network for the longitudinal prediction of future bone lesions for T1 weighted whole body MR images. The proposed cascaded architecture, consisting of two distinct configured U-Nets, first detects the bone regions and subsequently predicts lesions within bones in a patch based way. The algorithm provides a full volumetric risk score map for the identification of early signatures of emerging lesions and for visualising high risk locations. The prediction accuracy is evaluated on a longitudinal dataset of 63 multiple myeloma patients.

**Keywords:** Longitudinal prediction, asymmetric cascade U-Net, Multiple Myeloma, bone lesion assessment

## 1. Introduction

A key component in assessing the progression status of Multiple Myeloma (MM) is the identification of prime markers of the disease's advance during its prestage (smoldering Multiple Myeloma (sMM)). sMM is the most common disorder that leads to the malignant transformation of plasma cells and B-lymphocytes and in symptomatic MM to myeloma cells (Tosi). These further interact with bone marrow cells, which trigger the osteoclasts' activity, enhances bone resorption, and inhibits osteoblasts. Consequently, this leads from bone infiltration to destruction. Diffuse or focal bone infiltration starting during sMM are routinely imaged using whole body (wb) MRI (T1, T2) (Dimopoulos et al., 2015; Merz et al.,

Figure 1: Framework proposed for mapping future lesion risk in a whole body MRI

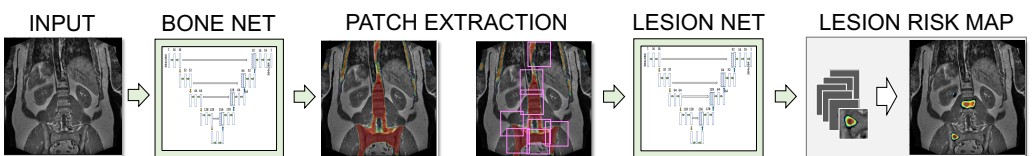

INPUT    BONE NET    PATCH EXTRACTION    LESION NET    LESION RISK MAP

2014; Kloth et al., 2014), while the gold standard to asses osseous destructions in late disease states is low-dose Computer Tomography (CT) (Lambert et al., 2017). The contribution of existing machine learning approaches lies in the detection of bone lesions. U-Nets have shown great potential in segmenting 2D microscopic images (Ronneberger et al., 2015), in detecting brain lesions (Kamnitsas et al., 2016) or bone lesions. In (Perkonigg et al., 2018) transfer learning is used to classify bone lesions in CT scans of MM patients. Our work was inspired by (Christ et al., 2017), where a cascade of two fully convolutional networks showed promising results for segmenting liver lesions in CT, by dividing the detection task in liver extraction and lesion detection within the liver region. Here, a method for the prediction of a bone lesion's future evolution risk in wbMRI is presented. This work provides the basis for effective treatment planning and response assessment during early MM stages, which shows a clear benefit for patients (Mateos et al., 2016). To our knowledge our method is the first approach providing a segmentation routine for bones in T1 wbMRI and also the first to perform longitudinal prediction of bone lesions in wbMRI using deep network architectures.

## 2. Contribution

The pipeline proposed consists of an asymmetric cascade of two U-Nets. Asymmetric, since a slice-based and patch-based U-Net (Fig. 1) are concatenated for slice-based bone segmentation (*BoneNet - BN*) and patch-based (*LesionNet - LN*) lesion prediction. Both U-Net architectures are based on (Ronneberger et al., 2015), but with exponential linear units in the convolution layers, Adam optimizer and a sigmoid function for the output. The proposed nets vary in terms of the dimensions of the layers and the loss function, where BN uses binary cross entropy (BCE) and LN a weighted BCE loss to overcome the imbalance between amount of lesion and non-lesion pixels. The BN's input is a 2D wbMRI slice (size $384 \times 384$ pixels) and the output a bone map. The LN's input are image patches extracted within the bone region of size $64 \times 64$ pixels (7k - 10k patches per slice, $\sim$200k per volume (30 slices)) and the output are patch-based lesion predictions which are further reconstructed to a full volume risk map. The longitudinal data used for training is preprocessed by first performing bias field correction and subsequently aligning follow-up images $(I_t, I_{t+1})$ of a patient by affine and non-rigid registration (Modat et al., 2014). The BN is trained using the dataset *Bone*:$\{I_t^p, B_t^p\}$, which consists of a patient's $p$ images and corresponding bone masks $B = B_t^1, \ldots, B_t^M$. The dataset *Lesion*:$\{I_t^p, S_{t+1}^p\}$ for training the LN are pairs consisting of an aligned intensity image of a subject at time point $t$ and aligned lesion annotations $A$ of the subsequent time point $t + 1$. The intensity images in *Lesion* are cropped using the corresponding thresholded (0.5) and dilated (2 pixels) bone maps $B_{t_i}$, which are also used to create patches out of the bone region with a sliding window approach.

Figure 2: Example of bone segmentation and lesion prediction results.

## 3. Results

The algorithm is evaluated on a data set of 220 longitudinal wbMRIs of 63 MM patients with overall 170 lesions. The data was acquired following (Durie et al., 2003) between 2004 and 2011. Bone and lesion annotations are provided by medical experts for every volume. In all experiments leave-one-out cross validation was used. We evaluated emerging lesions (non annotated in the input image but in the future state), for the thoracic/abdominal body part ($\text{BP}_{thorax}$) and in the pelvic/extremities body part ($\text{BP}_{legs}$). In the extremities' region a higher bone segmentation performance is achieved with a mean Area under the ROC (AUC) of 0.8023. Bone lesions are predicted with a mean AUC between 0.6083 ($\text{BP}_{thorax}$) and 0.5304 ($\text{BP}_{legs}$). The mapping of bone lesions in the future is a challenging task, since bone anomalies (not evolving towards lesions) can trigger false positives, while false negatives suggest weak early signatures. However, results show that the predicted risk score is capable of capturing early signatures of emerging lesions also illustrated in Figure 2. Here, a qualitative result for focal lesion prediction in $\text{BP}_{legs}$ is shown (from left to right): (1) input image $I$ at timepoint t, (2) detected bone mask (3) predicted lesion risk map (red=high risk, blue = low risk) (4) detailed view on the risk map and target image in an anomaly region ($1^{st}$ row) and lesion region ($2^{nd}$ row), (5) manual future lesion annotations (cyan) and corresponding target image at timepoint $t + 1$ (3 years later). The risk map at the distal part of the left femur visualises a high risk, which draws correspondence with the annotations of the future state. At the right distal femur, the proximal part of the right femur as well as at the right and left trochanter major, local anomalies in the bone are visible, which falsely are predicted as lesion, but do not progress to lesions in the future scan.

## 4. Conclusion

In this work we presented an asymmetric cascade network for the prediction of future focal bone lesion evolution for marking and assessing high risk bone regions. It consists of a slice-based *Bone Net* for the detection of bone in wbMRI and a subsequent patch-based *Lesion Net* for lesion prediction. This is the first attempt which is capable of predicting lesions on full volumetric wbMRI and demonstrated feasible results. We observed that anomalous bone regions are the main triggers for false positives predictions, which do not progress to lesions. In the future we plan to extend the approach by incorporating additional modalities to assess infiltration patterns during multiple myeloma progression.

## Acknowledgments

This work was supported by the DFG and the Austrian Science Fund (FWF) project number I2714-B31

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
