# OpenReview forum: "Asymmetric Cascade Networks for Focal Bone Lesion Prediction in Multiple Myeloma"
_MIDL.io/2019/Conference/Abstract — MIDL Abstract 2019_

### Official Review · AnonReviewer2 · 2019-04-26
**does not contribute a major breakthrough but shows an interesting application**

**Rating:** 3
**Confidence:** 3

**Review:**

This abstract shows how to establish a pipeline of convolutional networks for assessing bone lesions.

Methodologically this abstract does not contribute major breakthroughs. It is the first work that performs longitudinal prediction of bone lesions in wbMRI to the best of my knowledge. Clinical relevance could be highlighted a bit more. It would be interesting to see how consistent an ensemble-like mixing of different architectures would perform in the two segmentation stages, especially regarding the discussed failure case.


a 'Short Title' should be added.
'full convolutional networks' -> 'fully convolutional networks'
'The data was acquired following (Durie et al., 2003)between' -- space missing after the bracket

---

### Official Review · AnonReviewer1 · 2019-04-30
**Interesting work on prediction of future multiple myeloma lesions in MR**

**Rating:** 3
**Confidence:** 2

**Review:**

This work describes a method for prediction of future multiple myeloma lesions in whole body MR. The method uses two consecutive U-net-like architectures. The first segments bone from MR slices and the second one analyses patches from the segmented bone to detect predicts lesions. The training/evaluation of this second network is performed using longitudinal data, MR image and lesion reference from a later MR scan.

The paper is nicely written, the topic is interesting. Given the short format, only few details are provided so interpretation of the results is a bit difficult. I am not sure why e.g. bone detection is evaluated – do the authors mean bone segmentation?
It would also be nice to list sensitivity and false positive rate for the lesion prediction given the optimal threshold.

---

### Decision · Program_Chairs · 2019-05-06
**Acceptance Decision**

Accept